# A New Gold Rush: A Review of Current and Developing Diagnostic Tools for Urinary Tract Infections

**DOI:** 10.3390/diagnostics11030479

**Published:** 2021-03-09

**Authors:** Raymond Xu, Nicholas Deebel, Randy Casals, Rahul Dutta, Majid Mirzazadeh

**Affiliations:** Wake Forest Baptist Medical Center, Department of Urology, Medical Center Blvd, Winston-Salem, NC 27157, USA; rxu@wakehealth.edu (R.X.); ndeebel@wakehealth.edu (N.D.); rcasals@wakehealth.edu (R.C.); rdutta@wakehealth.edu (R.D.)

**Keywords:** UTI, urinary tract infection, urine culture, NGS, next generation sequencing, EQUC, expanded quantitative urine culture

## Abstract

Urinary tract infections (UTIs) are one of the most common infections in the United States and consequently are responsible for significant healthcare expenditure. The standard urine culture is the current gold standard for diagnosing urinary tract infections, however there are limitations of the test that directly contribute to increased healthcare costs. As a result, new and innovative techniques have been developed to address the inefficiencies of the current standard—it remains to be seen whether these tests should be performed adjunctly to, or perhaps even replace the urine culture. This review aims to analyze the advantages and disadvantages of the newer and emerging diagnostic techniques such as PCR, expanded quantitative urine culture (EQUC), and next generation sequencing (NGS).

## 1. Introduction 

Dr. Robert Koch was a German microbiologist and Nobel Laureate perhaps best known for his research and discovery of the Tubercle Bacille (TB) bacterium in the late 19th century. However, he is also often considered one of the “fathers of bacteriology” for propagating the germ theory of disease and developing four basic criteria to demonstrate that disease is caused by a particular organism, which is now known as Koch’s postulates. In his 1881 paper “Zurr Untersuchung von Pathogenen Organismen” (On the Examination of Pathogenic Organisms), Koch demonstrates why he is also referred to as the “grandfather of cloning,” as he describes a technique to grow isolated colonies of bacteria that would lay the foundation for one of the most commonly used laboratory techniques today: the cell culture [1].

While the application of cell culture in an investigational context is most associated with creating model systems to study basic cell biology, disease mechanisms, or novel drug toxicities, its use in a clinical context holds significant value as a diagnostic tool. The standard urine culture is a primary example as it is the current gold standard for diagnosing urinary tract infections (UTI’s), a leading cause of bacterial infections in the United States. UTI’s account for an estimated seven million office visits, one million emergency department visits, and over 100,000 hospitalizations annually [2,3]. As a result of its high incidence, UTI’s are responsible for $1.6 billion in annual healthcare costs, which represents a clear burden in the national healthcare system [4].

While it is a relatively reliable test, the standard urine culture (SUC) has many inefficiencies that contribute to overwhelming healthcare expenditure [5]. In a time where scientific/medical innovation is spurred by inefficiencies in clinical practice, new technologies such as PCR, expanded quantitative urine culture (EQUC), and Next Generation Sequencing (NGS) have arisen in hopes of revolutionizing UTI diagnostics. In this article we will review each of these diagnostic techniques, including the current gold standard urine culture, as well as future approaches for the diagnosis of the urinary tract infection. 

## 2. Standard Urine Culture with Sensitivity 

The standard urine culture (SUC) is one of the most widely used diagnostic tests in healthcare today. National surveys have showed that outpatient urine cultures are used in up to 77% of female patients with reported symptoms of urinary tract infection (UTI) (e.g., dysuria, urgency, frequency) [6]. The standard urine culture utilizes 1 mL of urine, typically obtained as a clean catch, midstream specimen (although catheterization can also be used), spread in pinwheel streaks onto 5% sheep blood (blood agar plate) and MacConkey agars, and aerobically incubated for 24 h (Figure 1).

Despite its prevalent use around the world, urine cultures are not typically warranted in healthy women with symptoms of UTI (i.e., uncomplicated UTI) because of the effectiveness of empiric antibiotic treatment [8]. The urine culture has minimal clinical value in this patient population due to inconveniences of time and cost for the test, and instead patients are diagnosed and treated based on symptoms. The utility of the urine culture is therefore reserved for inpatient and complicated UTI’s (e.g., refractory to initial treatment, pyelonephritis, asymptomatic bacteruria of pregnancy, atypical organisms, etc.) to properly diagnose and treat the infection [9]. The emergence of antibiotic resistance has elevated the urine culture to be the gold standard diagnostic test, as its ability to identify the responsible pathogen and its antimicrobial sensitivities allows for responsible practice and antibiotic stewardship. This is especially useful in complicated and recurrent UTI’s in which successful eradication of the pathogen is necessary to prevent injurious sequelae (e.g., upper urinary tract damage). 

The urine culture also allows for an objective measure for UTI diagnosis through quantification, however the thresholds for what constitutes a UTI vary depending on procurement method. The diagnosis of a UTI from a clean catch specimen therefore, as determined by the Centers for Disease Control and Prevention, includes symptoms plus 10^5^ colony forming units (CFU)/mL. The threshold for UTI in a catheterized specimen is then 10^3^ CFU/mL. Historically, the urine culture has been a highly reliable test, with a reported 95% sensitivity and 85% specificity based on thresholds as low as 10^2^ CFU/mL [10]. Needless to say, the urine culture is a valid diagnostic tool, however it is not without its share of inefficiencies. 

Despite being the current gold standard for UTI diagnosis, the SUC has several shortcomings. For one, cultures are time consuming and take a minimum of two days to receive a result with corresponding sensitivity profiles. This creates a delay in treatment that opens an opportunity to develop unwanted complications such as upper urinary tract involvement (e.g., pyelonephritis), an unnecessary risk particularly if the symptoms can be quickly treated with empiric antibiotics. 

While the SUC has shown to be a reliable test for diagnosing UTI’s caused by *Escherichia coli* (80–85% of all UTI’s) [11], its ability to discriminate the causative organism from other uropathogens (e.g., gram-positive bacteria; *Enterococci* and *Group B streptococci*) has called into question its reliability. Hooten et al. demonstrated that the positive predictive value of midstream urine cultures was 93% and 99% for *Escherichia coli* growth of at least 10^2^ CFU and 10^4^ CFU, respectively, but the positive predictive value was 10% and 33% for En*terococci* growth and 8 and 14% for *Group B streptococci* growth of at least 10^2^ CFU and 10^4^ CFU, respectively [12]. Furthermore, in this study the presence of *Enterococci* and *Group B streptococci* were frequently found in cultures from midstream urine, but not from catheter urine specimens taken from women with cystitis, suggesting potential for false positive reporting of *Enterococci* and *Group B streptococci* cultures from the standard midstream urine collection. These results can lead to misinterpretation and inappropriate treatment, contributing to the pervading problem of antimicrobial resistance [13].

The SUC is also highly susceptible to contamination, with an average reported rate of 15% at institutions across the nation [14]. Contamination is likely when urine culture results in low bacterial growth and/or growth of several bacterial species—*Lactobacilli*, *Corynebacterial* spp., *Gardnerella*, *alpha-hemolytic Streptococci*, and aerobes are considered urethral and vaginal contaminants. A meta-analysis done by Larocco et al. demonstrated that contamination often occurs during the pre-analytic phase of urine culture, and can be reduced with proper techniques for urine collection, storage, and transport [15]. Furthermore, the false positive results created by contamination leads to suboptimal or unnecessary treatment, leading to poor patient outcomes and added healthcare costs.

Another major drawback of the standard urine culture is the inconsistency in threshold definitions for which different institutions consider clinically significant [16]. While the majority of clinical microbiology laboratories in the United States deem a colony count of more than 10^5^ colony forming units (CFU)/mL to be diagnostic of a clinically significant test result, there is an argument to be made that this cut off is too high and may result in overlooking more insidious infections [10]. Even more liberal thresholds of 10^3^ CFU/mL can fail to detect some sexually transmitted infections [17]. In fact, it has been reported that 20% to 40% of women with symptomatic UTI’s present with bacterial counts of 10^2^ to 10^4^ CFU/mL of urine [18]. For this reason, Stamm and Hooten suggested a threshold as low as 10^2^ CFU/mL for dysuric patients [3]. The lack of standardization for what qualifies as a UTI is even more evident when considering that some microbiology laboratories include additional charges for routine and comprehensive culture tests. A routine urine culture test costs about $34, whereas a comprehensive urine culture test costs nearly double that. While both tests include culture, quantification, as well as identification and susceptibility testing of detected organisms, the comprehensive test is far more sensitive with a threshold of 10^2^ CFU/mL (the routine culture has a threshold of 10^4^). The lack of consensus for what constitutes an infection, combined with high rates of contamination, has called into question the reliability of the current gold standard urine culture.

## 3. Enhanced Culture for Atypical Organisms

The standard urine culture media is selective for typical microorganisms most commonly responsible for urinary tract infections, including *Escherichia coli*, *Staphylococcus Saprophyticus*, *Klebsiella pneumoniae*, and *Proteus mirabilis*. Therefore, it is not an applicable test when there is a high clinical suspicion for UTI caused by atypical organisms, as seen in neutropenia (e.g., candiduria), genitourinary tuberculosis, and urinary tract abnormalities (e.g., anaerobic bacteria). These situations necessitate the use of augmented culture methods and diagnostic tests, which vary by microorganism. 

Candiduria, the presence of *Candida* spp. in the urine, is the most common fungal etiology of nosocomial UTI’s. While it is commonly considered a benign and asymptomatic condition, there are several large studies that report increased mortality of candiduric patients compared to controls [19,20]. Candiduria may either represent colonization or actual infection, the latter of which may be indicative of invasive candidiasis and requires aggressive treatment to prevent increased mortality and medical burden [21]. A systematic review done by Nickel demonstrated that *Candida* cultures tend to be representative of colonization when associated with a catheter, particularly in hospitalized patients already on broad-spectrum antibiotics; however, invasive candidiasis was more likely in ICU and neutropenic patients [22]. Given that the incidence of UTI’s secondary to candiduria may be increasing, accompanied by the ever-present concern of antifungal resistance, the urine culture becomes essential to monitor the susceptibility profiles of *Candida* species causing candiduria [23]. Toner et al. demonstrated that the prevalence of candiduria in a contemporary European database is 2.6 per 1000 positive urine cultures, and the majority of cultures positive for *Candida* spp. were due to *Candida albicans* [24]. Furthermore, 94% of *Candida albicans* were susceptible to fluconazole, and 100% were susceptible to amphotericin B treatment. The utility of fungal culture and sensitivity is therefore to monitor the efficacy of current antifungal treatment guidelines for candiduria. 

While *Candida* spp. can be isolated from the standard urine culture (blood and MacConkey agars), a fungal culture medium (Sabouraud dextrose agar) may be used to better detect candiduria. Helbig et al. performed a comparison of recovery by standard and fungal urine cultures, using samples that demonstrated presence of yeast on urinalysis—the standard urine culture detected only 37% of *Candida* spp., while the fungal culture medium recovered 98% of *Candida* spp. [25]. Taken together, these results indicate that candiduria is not adequately identified by standard urine culture methods.

While uncommon in the United States, genitourinary tuberculosis (TB) is the third most common presentation of extrapulmonary tuberculosis worldwide, behind pleural and lymphatic TB. Patients typically present with non-specific symptoms, which often leads to difficulty and delays in diagnosis. The current gold standard for diagnosing genitourinary TB is augmented urine culture (Lowenstein-Jensen medium), which requires up to 8 weeks to recover *Mycobacterium tuberculosis* and has a false negative rate of up to 20% [26]. Due to the insidious onset of symptoms, diagnosis and treatment of genitourinary TB is often delayed, resulting in significant morbidity or even mortality. Diagnosis can also be made with nucleic amplification tests, which may be a more sensitive method of diagnosis.

The omission of anaerobic bacteria (e.g., *Lactobacillus* spp., *Clostridium* spp., *Bacteroides* spp., *Peptostreptococcus* spp., *Peptococcus* spp.) from routine urine culture testing is due to the low pathogenicity within the urinary tract, making it an improbable cause of UTI [27]. Anaerobic bacteria are part of the normal flora of the vagina, large intestine, and skin, and do not typically cause UTI except in patients with urinary tract abnormalities (e.g., enterovesicular fistula, traumatic urinary tract injury, pelvic infection, etc.) [28]. Therefore, utilization of anaerobic-specific cultures should be considered in patients who are at enhanced risk, and also in those with symptoms of UTI and presence of bacteria on microscopic evaluation with negative urine cultures.

*Ureaplasma* spp. and *Mycoplasma hominis* are two fastidious bacteria (Mollicutes class) that are associated with UTI’s. They are exceptionally difficult to cultivate in standard urine culture due to their lack of cell walls, and require special inoculation on A7 agar, which directly tests for presence of urease, allowing differentiation of *Ureaplasma* spp. from other *Mycoplasmatales* [29]. While the A7 agar is specific (by definition 100% specificity for *Ureaplasma* spp. and *Mycoplasma* spp.), studies have demonstrated sensitivities of 70% [30]. As with all culture methods it requires several days for detection, which has led to the investigation of more rapid tests such as multiplex PCR. However, the utility of identifying antimicrobial resistance remains an important factor in urine culture testing [31].

## 4. Polymerase Chain Reaction (PCR)

Polymerase chain reaction (PCR) based diagnostic testing is one of the many advances in DNA-related laboratory techniques that has become widely available in a variety of settings. The introduction of multiplex PCR testing, which utilizes multiple primers to detect several targets at once has dramatically reduced the cost and time associated with this class of test as well as increasing its utility in clinical medicine [32]. It is particularly useful for identifying infectious agents, and has been studied for several types of infections including but not limited to bloodstream infections, sexually transmitted infections, and gastrointestinal infections [33]. The appeal of PCR-based diagnostic testing for UTI’s lies in its high specificity and sensitivity as well as the rapidity with which results can be obtained compared to standard bacterial cultures for the previously named infections [34,35,36], among others. PCR not only has higher detection rates of single pathogens compared to urine culture [37,38,39], but has been shown to be effective in detecting multiple pathogens in urine, which standard urine cultures routinely fail to do [33]. This is of great importance considering that between 30–39% of UTI’s are polymicrobial [33,40,41,42] and appropriate antimicrobial treatment is dependent on accurate identification of the causative organisms. Antibiotic resistance in UTI’s is seen in both inpatient and outpatient cases of infection [43] and targeted treatment is an important facet of antibiotic stewardship to prevent the continued propagation of resistant pathogens. 

Currently, the standard of care for UTI’s is to begin empiric antimicrobial therapy until urine culture results are available, which is typically at least 48 h after collection [44]. In a proof-of-concept experiment, a commercially available qualitative PCR assay of common bloodstream pathogens was shown to identify urinary pathogens at least 43 h sooner and with a sensitivity and specificity of 80% and 60%, respectively, compared to urine culture [45]. In a separate study, urine PCR surpassed the investigators’ established non-inferiority threshold and had 90% agreement with urine culture results with the added benefit of faster results as well as improved detection of polymicrobial infections [38]. Decreasing the amount of time that a patient receives empiric treatment before the pathogen is identified may improve patient outcomes by limiting exposure to the adverse effects of empiric treatment as well as shortening the duration of time that patients require inpatient treatment, which decreases their risk of nosocomial infection. The relative ease with which PCR testing of urine can be automated [32], in addition to its accuracy as a diagnostic tool and the speed with which results are available make PCR-based urine testing for UTI’s a formidable addition to the physician’s armamentarium. 

In spite of the benefits of PCR-based diagnostic testing, it cannot as of yet fully replace the utility of the urine culture. Although PCR can uniformly provide faster identification of pathogens, it is also subject to identifying pathogens that are either present in inconsequential amounts in the urine or a part of the patient’s urinary microbiome [38]. Additionally, while identification of the causative organism helps tailor antimicrobial treatment, most PCR results are unable to provide information regarding sensitivity of the pathogens to different treatments, unlike the standard urine culture [46]. Currently, the detection of antibiotic resistance in bacteria present in urine is limited to a handful of known resistance genes [46], or a more robust investigation of several treatment resistance-associated genes in multiplex PCR testing of a single family of bacteria such as *Enterobacteriaceae* [47]. Schmidt et al., suggest that this admittedly limited capability of PCR testing may still offer clinicians the ability to guide treatment decisions early in the disease course by including genes commonly associated with drug-resistance in urinary pathogens. These include genes that confer resistance to trimethoprim, aminoglycosides, and fluoroquinolones, in addition to genes for extended-spectrum β-lactamases, *ampC*, and carbapenemases [47]. The authors argue that it is a tool for use in conjunction with, and not a replacement for, urine cultures that can confirm that pathogens do not harbor rare mechanisms for resistance that are untested, but will likely offer an early start to appropriate treatment in most cases. Such an approach has yielded positive results with tuberculosis [48] and has the potential for utility in the treatment of UTI’s in both inpatient and outpatient settings [47].

PCR testing is also limited by higher costs and the capability of laboratories to perform the tests compared to the lower technical capabilities required to interpret urine cultures [32]. With PCR testing as it stands now, these costs would be added on to the existing costs of urine cultures which would need to be performed in tandem to confirm pathogen treatment sensitivities. It has yet to be determined whether advanced multiplex PCR testing with the capability to provide a more complete picture of antibiotic resistance of the identified pathogens as well as increasing access to the necessary technology will lead to enough cost-savings to outweigh the current financial and logistic impediments. 

## 5. Expanded Quantitative Urine Culture (EQUC)

Expanded quantitative urine culture (EQUC) is a variation of the urine culture that detects live microorganisms in urine specimens that are not detected by the standard protocol. Wolfe and colleagues were the first to describe the presence of “uncultivated” bacteria in the bladders of healthy females without symptoms suggestive of urinary tract infection (UTI) by using 16S rRNA sequencing, light microscopy, and PCR [49]. While it was unclear whether or not the observed bacteria were viable, the lab’s findings suggested that urine within the bladder may not be sterile even in healthy individuals. Two years later, in 2014, they described the first protocol for an EQUC [41] that utilized a larger volume of urine (10–100 mL instead of 1 mL for a standard culture), multiple growth media (5% sheep blood/blood agar plate, MacConkey agar, chocolate agar, colistin-nalidixic acid agar), longer incubation times (up to 48 h), and a variety of atmospheric conditions to cultivate bacteria that may not grow on a standard urine culture. With this, bacterial presence as low as 10 colony forming units (CFU) per mL could be detected; indeed, *Lactobacillus*, *Corynebacterium*, and multiple other genera were isolated using this EQUC from the bladders of women with and without overactive bladder (OAB) syndrome symptoms. 92% of those urine samples that yielded bacterial growth with the EQUC showed none on standard urine culture [41]. They proposed a “streamlined” version of this EQUC, which specified using a higher volume of 100 mL of urine on MacConkey, blood, and colisitin-nalidixic acid (CNA) agars in a 5% CO_2_ incubator for 48 h to yield 84% sensitivity relative to the extended spectrum protocol. This protocol has been utilized by several investigators since. 

While it is now established that the bladder has its own unique microbial community in both healthy individuals and those with urologic pathology, the exact clinical relevance of these findings remains unclear [50]. In a study of 150 women, Price and colleagues grouped patients by whether or not they had self-reported UTI-like symptoms and performed the aforementioned streamlined EQUC on catheterized urine specimens; while they did not find a difference in the number of isolated uropathogens, they did find a reduced species richness and diversity in patients who did have clinical UTI symptoms [40]. About half of the uropathogens in the UTI cohort were missed by standard urine culture; additionally, the threshold of 10^5^ CFU/mL would not report a predominant organism in numerous patients with a clinical UTI in this cohort. Another study of 570 pediatric patients at a hospital in Nepal with symptoms concerning for a UTI found that an EQUC picked up significantly more known uropathogenic microbes than standard culture, including *Candida albicans*, *Provedencia retergerii*, and *Morganella morganii* [51]. In the same study, EQUC also identified all probable uropathogens including extended-spectrum beta-lactamase (ESBL) producers, multi-drug resistance (MDR) and extensive drug-resistant (XDR) organisms that would be overlooked by SUC [51]. While these studies show promise in improved detection of uropathogens, including those that may be MDR or XDR, future studies following the treatment and potential symptomatic resolution in patients with previously “uncultivated” microbes are necessary.

Data regarding the urinary microbiome and interstitial cystitis (IC), a condition classically characterized by lower urinary tract symptoms (LUTS) such as dysuria, frequency, and urgency in the absence of a positive urine culture, are conflicting. In a study of 49 women with IC and 40 control patients, the authors found a slightly higher proportion of identifiable microbes in the catheterized urine of IC patients; however, within the IC cohort itself, there was no association between symptom severity and bacterial abundance. The authors even noted that patients who were symptom-free within the IC cohort were more likely to have a positive EQUC [52]. A similar, but smaller study utilizing mid-stream urine collections yielded similar results [53]. 

EQUC techniques have been utilized to study a variety of other disease states. In a study of exclusively male patients undergoing surgical therapy for LUTS (mostly attributed to benign prostatic enlargement) and controls undergoing another surgical procedure, both mid-stream and catheterized urine samples were obtained [54]. While nearly all of the mid-stream specimens had growth by EQUC, only 39% of catheterized samples did, implying that a significant microbial burden in urine specimens may arise from the urethra and urethral meatus in men. Additionally, there was a significant association between LUTS symptom score category and the likelihood of identifying uropathogenic bacteria in catheterized specimens, implying that male LUTS previously attributed entirely to anatomic obstruction may be partially related to the urinary microbiome makeup. In a study of women with and without urge urinary incontinence (UUI), EQUC of catheterized samples showed an increased median number of bacterial isolates from women with UUI [55]. In addition to this, they found that the species of *Lactobacillus* differed between groups while the genus remained dominant, and the UUI group had more bacteria from the genera *Corynebacterium* and *Streptococcus*. Another study of women with UUI utilizing EQUC corroborated this finding of both more bacteria and more microbial diversity in women with UUI [56]. In this study, women with UUI were subsequently treated with solifenacin, an anticholinergic; women who responded to the treatment had fewer bacteria and less microbial diversity than those who did not. Another small, though interesting study performed EQUC on both urinary stones (primarily calcium-based) and the adjacent urine (whether from bladder or upper tracts) and found that certain species appeared in both samples but were more “dominant” within the stone culture, indicating that certain bacterial species may be more extensively incorporated in urinary stones [57]. While a great deal of data has been generated using EQUC, the clinical implications remain to be seen. Future studies regarding the pathogenicity of various microbes, unique CFU thresholds, and the responses to treatment are warranted.

## 6. Next Generation Sequencing (NGS)

The use of NGS is currently considered to be the forefront of diagnostics in UTI’s. It was previously believed that human urine was sterile and any deviation from this was considered abnormal. This theory was later called into question by the initiation of the National Institutes of Health sponsored Human microbiome project which demonstrated the presence of a bladder and urinary microbiome [56]. While the data surrounding the urinary microbiome is still in its infancy, existing work has shown the primary occupants to include organisms from the genera *Lactobacillus*, *Garderella*, *Streptococcus*, *Staphylococcus* and *Corynebacterium* [58]. 

NGS is culture independent and does not require the prolonged growth of organisms. Instead, it utilizes the PCR and high throughput sequencing of the 16S ribosomal RNA (rRNA) gene (Figure 2) [56]. This gene is frequently used as it is highly conserved amongst bacteria given its essential function. However, within the 16S rRNA gene there are 9 hypervariable regions that are recognized to have polymorphisms which allow for further speciation [59]. When a sample of urine is tested using NGS, the output reports bacterial presence in terms of relative sequence reads amongst species instead of absolute values [60]. To date, MicroGen DX and Aperiomics are two companies which offer NGS as a clinical application for the diagnosis of UTI’s [58]. 

There is a paucity of data examining the use of NGS as a diagnostic modality for urinary tract infections. However, several studies have shown the potential utility of NGS given its enhanced ability to detect organisms missed by standard urine culture. In a study by McDonald et al. 44 patients were treated for their UTI based on culture and sensitivity (control) or NGS results [61]. Additionally, all patients also completed the UTI symptom assessment questionnaire prior to and after completion of antibiotic treatment. Overall, 13 of 44 patients (30%) were positive for UTI on urine culture compared to the same 44 patients who were positive for UTI according to NGS results. Additionally, patients treated based on NGS results alone had a significant reduction in symptom severity in comparison to patients that were treated based on urine culture alone. 

In a multi-institutional study by Pearce et al., the urinary microbiome of females with UUI was examined [55] As part of the ABC trial, these patients were screened for UTI via urinalysis of a catheterized specimen and were all found to be leukocyte esterase and nitrite negative. Each patient’s urine sample was additionally sent for NGS, which found that approximately 51% (n = 93) of patients were sequence-positive despite a negative UA [55]. Of note, the predominant organisms detected with NGS included: *Lactobacillus*, *Gardenerella*, *Gardenerella/Prevotella*, *Enterobacteriaceae*, *Staphylococcus*, *Aerococcus*, and *Bifidobacterium*. 

Finally, in an observational, retrospective case series by Ishihara et al., traditional urine/blood cultures and NGS were used study the urine and blood of 10 patients presenting to the emergency department with acute urinary tract infection, 4 of which were classified as having urosepsis [62]. Overall, 90% of patients were pathogen-positive on urine culture compared to 100% of patients identified as pathogen-positive by NGS. The authors further stated that the organism identified on culture was also the predominant organism expressed in the NGS output. However, as in previous studies, the authors noted that many other organisms with pathogenic potential were identified on NGS alone. While underpowered, this study helps to confirm that NGS may offer additional information regarding unculturable organisms with pathogenic potential. The results of this study were similarly echoed by Lewis et al. who examined NGS results from mid-stream urine samples of 16 patients [5,63]. They were able to demonstrate the presence of 94 bacterial genera within the samples. The authors state that only 31 of these would be routinely cultivated or separately identified by an NHS microbiology laboratory. They additionally identified the presence of the anaerobic genus *Soehngenia* for the first time in human urine [63]. 

To date, the literature implies that NGS has several positive attributes to contribute to the diagnostic armamentarium of urinary tract infections. First, it is a highly accurate test with reported 100% sensitivity and 95.5% specificity for bacteria and 100% sensitivity and 97.3% specificity for fungi per MicroGen’s website. NGS also has the potential to be a cost-effective tool relative to the urinary culture. Currently, 16S rRNA sequencing through the MicroGen Dx DNA sequencing platform is approximately 200 dollars [63]. In comparison, the cost of a urine culture can range from 60–500 dollars depending on the extent of testing for atypical species [58,61]. However, as previously demonstrated, standard urine culture may fail to identify some organisms even when testing for fungal or anaerobic organisms is performed. To this end, NGS is superior as they utilize two separate PCR panels that differ in primer binding sites and technology. Level I reports are more sensitive as they are able to detect panel organisms at a lower concentration than level II, however level II reports will detect many more organisms including those that have had a strain mutation (Figure 3). Therefore, there are several “difficult to treat” patient populations in which NGS may prove to be a better diagnostic tool. Table A1 demonstrates several focused patient populations in which the use of NGS may hold clinical benefit. 

Despite the promise of NGS in urinary diagnostics, NGS has its apparent limitations. As previously demonstrated, NGS has the capacity to identify numerous organisms that traditional urine culture does not. However, given the shift in paradigm regarding the presence of a urinary microbiome, questions about the clinical significance of these organisms remains poorly defined. While the presence of organisms are reported as a proportion relative to others, this does not definitively translate to identification of a causative organism. As stated by Dixon et al., future work is needed in order to identify prediction algorithms which would aid in NGS output analysis and targeted treatment [64]. This will also help to prevent overtreatment of organisms that may be part of the “normal” urinary microbiome. 

Another serious limitation of NGS technology is the quality of the genomic reference libraries. In their current state, the databases are public in nature and can be annotated by anyone. Unfortunately, this leads to annotations of sequences and genomes that are incorrect [5,64]. Without accurate reference materials, it is difficult to trust the results of NGS outputs being used to make clinical decisions outside the context of a clinical trial. Therefore, as stated by Dixon et al., it is likely that in order for NGS to be used clinically, tighter regulations over genomic reference libraries will be needed in order to ensure quality control [5].

## 7. Discussion

In this review we examined the advantages and disadvantages of all current diagnostic tools for UTI’s (Table 1). The urine culture is the gold standard for diagnosing urinary tract infections, a leading cause of bacterial infections in the US. While it is a reliable test that is able to identify the causative organism as well as sensitivities to various treatment options, it is marred by several shortcomings including at least a 48 h wait for final results, susceptibility to contamination, and reduced reliability for less common uropathogens. Additionally, while the standard urine culture is effective for common causes of UTI’s, additional methods are required to cultivate atypical uropathogens including fungal agents. PCR diagnostic testing of urine improves on the standard urine culture by drastically reducing the average time for results from several days to several hours, as well as by offering increased utility for polymicrobial infections. However, while PCR is able to detect common genes associated with treatment resistance, it is considerably less effective at providing physicians with definitive data on treatment resistance. EQUC, requiring a larger volume of urine than standard culture and the utilization of several growth media and conditions, offers improved microbial identification but has limited data available regarding its feasibility for regular clinical use. Finally, NGS utilizes PCR and high throughput sequencing in order to provide comprehensive data on uropathogens in a patient sample with increased detection of pathogens routinely missed on standard culture at a comparable price. While NGS is surely the most accurate and sensitive test available, it remains to be seen whether the wealth of data it provides (e.g., identification of all bacteria within a urine sample) has significant clinical relevance, as it may not be able to distinguish uropathogens from the normal urobiome.

## 8. Conclusions

The standard urine culture as has remained the gold standard of UTI diagnosis for good reason. It offers the most reliable data to form a treatment plan in a reasonable time frame for the most common pathogens. However, as medical technology continues to advance, newer diagnostic tools such as PCR, EQUC, and NGS provide more comprehensive data on pathogens and their sensitivities to treatment in less time. It is likely that the next generation of physicians will routinely reach for one of these or similar tests to accurately diagnose a suspected UTI and formulate a treatment plan within the time constraints of the average clinic visit.

## Figures and Tables

**Figure 1 diagnostics-11-00479-f001:**
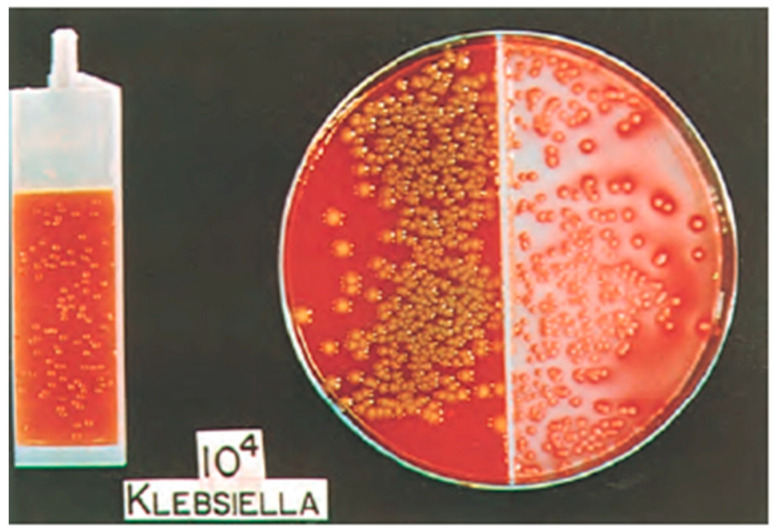
Split-Agar Urine Culture—blood agar (**left**) and eosin-methylene blue (EMB) (**right**). Adapted from Campbell-Walsh Urology 11th edition [7].

**Figure 2 diagnostics-11-00479-f002:**
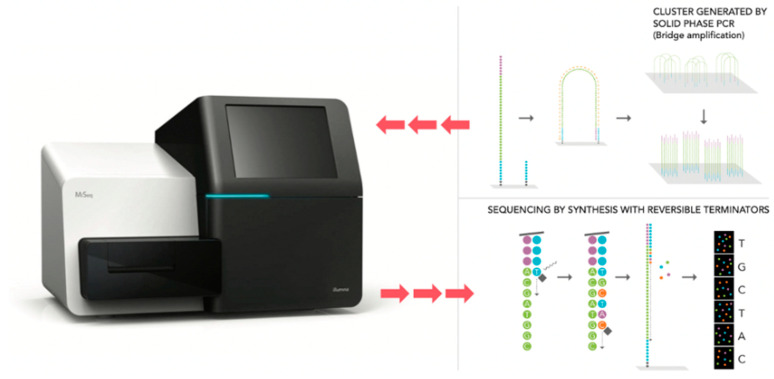
Next Generation Sequencing. Microbial DNA is detected and amplified with bridge PCR for high-throughput sequencings.

**Figure 3 diagnostics-11-00479-f003:**
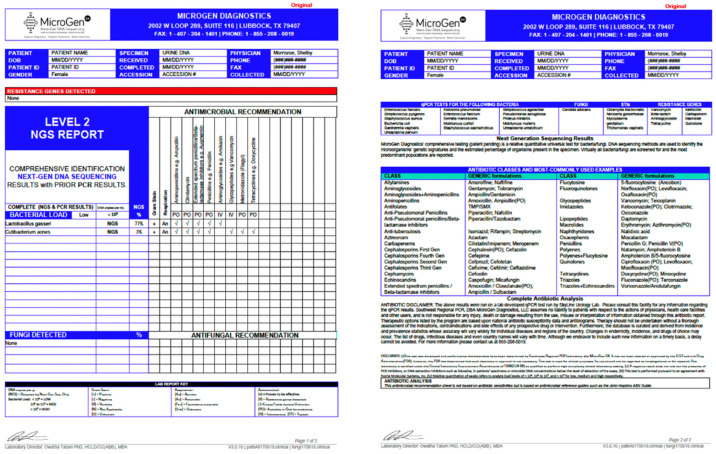
MicroGen DX Rapid Screening Results Sample Report.

**Table 1 diagnostics-11-00479-t001:** Comparison of Different Diagnostic Methods of Urinary Tract Infections.

	SUC	EQUC	PCR	NGS
**Diagnostic Tool**	Microbiology; developed in 1880’s, designed for acute infections	Advanced Microbiology; developed in 2014, designed for acute infections	Molecular Method; developed in 1970’s	Advanced Molecular Method; developed in 2004
**Methodology**	Detects species by growing bacteria or fungi on petri or agar plates	Detects atypical and subthreshold species by growing bacteria or fungi on petri or agar plates under modified conditions	Matches extracted microbial DNA to limited PCR panels	Matches extracted microbial DNA to large curated species libraries
**Ability to Detect Dominant Species**	Can identify dominant species with high sensitivity for common uropathogens	Identification of multiple isolates may represent contamination (i.e., requires catheter-collected urine for accuracy)	Cannot reliably discern uropathogen from normal uromicrobiome	Identifies all species in a sample and lists by dominance
**Antibiotic Sensitivities**	Reliable identification of antibiotic sensitivity profiles	Reliable identification of antibiotic sensitivity profiles	Inconsistent inclusion of resistance genes in PCR panels	Dependent on public genomic reference libraries for resistance genes results
**Specimen Management**	Sensitive to time and temperature	Sensitive to time and temperature	Not easily affected by time or temperature	Not easily affected by time or temperature
**Turn-Around Time**	1–2 days for bacteria; 20+ days for fungi	1–2 days for bacteria; 20+ days for fungi	1 day	3–5 days
**Costs**	−$34 −$71 (fungal) −$114 (acid fast) -Variable (anaerobe)	−$67	−$15–30	−$200

## Data Availability

No new data were created or analyzed in this study. Data sharing is not applicable to this article.

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
