# Peer review of "A New Gold Rush: A Review of Current and Developing Diagnostic Tools for Urinary Tract Infections"

_diagnostics, 2021, doi:10.3390/diagnostics11030479_

Round 1

Reviewer 1 Report

Dear Authors,

This narrative review presents the current and developing modalities of diagnosis UTIs, the current role of urine culture, and it is interesting for all medical and surgical physicians. 

The manuscript is well written, organized, and structured. Here are my comments:

Major aspects:

Please include in the EQUC section a discussion regarding the detection of MDR and XDR uropathogens.

Minor aspects:

  • please clarify what EQUC means expanded or enhanced and verify the whole manuscript
  • urinary tract infections - UTIs (please check the whole manuscript for consistency of this abbreviation)
  • l. 47 should be Standard Urine Culture (SUC)
  • l. 81 should be 2 days (not 3 or 4 days) - see l. 208
  • l. 89 -92 it is not clear the relevance; the standard is 105 or 10from catheter
  • l. 211 should be 80% and 60%?
  • fig 3 should be supplementary material
  • l. 393 Table 2

l. 425 the authors should change Discussion with Clinical Applications or Current Diagnostics Tools for UTIs

l.437 the authors should insert Conclusions section

The authors should include table 1 in the main text. 

Author Response

Reviewer 1 Comments & Author Responses

Point1: Please include in the EQUC section a discussion regarding the detection of MDR and XDR uropathogens.

Response 1: This has been added beginning with l. 285.

Point 2: Please clarify what EQUC means expanded or enhanced and verify the whole manuscript.

Response 2: We changed all “enhanced” to “expanded”

Point 3: Urinary tract infections - UTIs (please check the whole manuscript for consistency of this abbreviation)

Response 3: all “UTIs” abbreviations were changed to “UTI’s”

Point 4: l. 47 should be Standard Urine Culture (SUC)

Response 4: This was changed.

Point 5: l. 81 should be 2 days (not 3 or 4 days) - see l. 208

Response 5: This was changed to “takes a minimum of 2 days to receive a result…”

Point 6: l. 89 -92 it is not clear the relevance; the standard is 105 or 103 from catheter

Response 6: This paragraph was not so much about the values at 105 or 103 CFU, but rather to demonstrate that the urine culture was not accurate at detecting other uropathogens such as enterococci and GBS.

Point 7: l. 211 should be 80% and 60%?

Response 7: This was changed.

Point 8: fig 3 should be supplementary material

Response 8: We specifically discuss the difference between “level I and level II” reports, which makes the image necessary in the body of the text for visual understanding.

Point 9: l. 393 Table 2

Response 9: This was changed.

Point 10: l. 425 the authors should change Discussion with Clinical Applications or Current Diagnostics Tools for UTIs

Response 10: Beginning with l. 415, we are summarizing each of the reviewed diagnostic methods, precluding the conclusion.

Point 11: l.437 the authors should insert Conclusions section

Response 11: A conclusions section was inserted.

Point 12: The authors should include table 1 in the main text.

Response 12: This was changed.

Reviewer 2 Report

Dear authors,

I found your work very interesting. My only point is that I would appreciate a paragraph on methods developed and published in literature in recent years but not yet commercialized (as the title refers to: developing diagnostic tools). In particular there are some methods that even if do not lead to the identification of pathogen, are quantitative, really not time consuming (few minutes), cheap (few tens of dollars) and with the potential to give information about antibiotic susceptibility. Method that, if used as early screening, have the ability to guide treatment decision, avoiding incorrect or unnecessary prescriptions of antibiotics.

Since antibiotic resistance is a problem of global concern, which could lead to another global pandemic, the focal point to evaluate different UTI's method in my opinion should be to obtain correct medical prescription and therapies.

Author Response

Thank you for the comments, can you provide some more details about which of these developing methods you are referring to? Our initial goal was to only review the established diagnostic tools that are currently being used.

Reviewer 3 Report

I found the paper of utmost importance. There is a need of novel tools in diagnostic of UTI, beyond urine culture. I do not have any remarks. 

Author Response

Reviewer had no remarks. Thanks